# Similar Quality of Life and Safety in Patients Receiving Inpatient or Outpatient Chemotherapy: A Focus on Esophageal Squamous Cell Carcinoma

**DOI:** 10.3390/healthcare8040447

**Published:** 2020-11-01

**Authors:** Yen-Hao Chen, Su-Wei Chen, Hung-I Lu, Chien-Ming Lo, Shau-Hsuan Li

**Affiliations:** 1Department of Hematology-Oncology, Kaohsiung Chang Gung Memorial Hospital and Chang Gung University College of Medicine, Kaohsiung 833, Taiwan; alex2999@cgmh.org.tw; 2School of Medicine, Chung Shan Medical University, Taichung 402, Taiwan; 3Department of Nursing, Meiho University, Pingtung 912, Taiwan; 4Department of Nursing, Chang Jung Christian University, Tainan 711, Taiwan; mimiko700531@gmail.com; 5Department of Anesthesia, Tainan Municipal An-Nan Hospital, Tainan 709, Taiwan; 6Department of Thoracic & Cardiovascular Surgery, Kaohsiung Chang Gung Memorial Hospital and Chang Gung University College of Medicine, Kaohsiung 833, Taiwan; luhungi@yahoo.com.tw (H.-I.L.); t123207424@cgmh.org.tw (C.-M.L.)

**Keywords:** quality of life, esophageal cancer, inpatient chemotherapy, outpatient chemotherapy

## Abstract

Quality of life (QoL) is a particularly important issue for cancer patients. This study was designed to investigate the differences in QoL in esophageal squamous cell carcinoma (ESCC) patients who underwent inpatient chemotherapy (IPCT) or outpatient chemotherapy (OPCT). A total of 107 ESCC patients were enrolled, including 53 patients in the IPCT group and 54 patients in the OPCT group. The widely used and well-validated instruments European Organization for Research and Treatment of Cancer Quality of Life Questionnaire Core 30 Items (EORTC QLQ-C30) and Oesophageal Cancer Module (EORTC QLQ-OES18) were used to examine the QoL of the two groups. In addition, the differences in adverse events (AEs) were evaluated. The results of QLQ C-30 analysis showed that mean global quality of life scores were similar between IPCT and OPCT groups, as were functional and symptom scales. There were no significant differences in the functional and symptom scales in the analysis of QLQ OES18 either. Most AEs of chemotherapy were grades 1–2, and the majority of patients tolerated the side effects; no statistically significant difference in AEs between these two groups was mentioned. Our study suggests that the health-related QoL and adverse events in ESCC patients who received IPCT or OPCT are similar. OPCT is reasonable and safe in clinical practice.

## 1. Introduction

Quality of life (QoL) is very important to individuals, and it may be affected by health and illness. The development of novel cancer treatment regimens has improved clinical outcomes, such as response rate and overall survival, but treatment-related toxicities have still been difficult to quantify [1]. Growing evidence has shown that the goal of cancer treatment should incorporate concerns not only regarding the efficacy of tumor control but also patients’ QoL throughout the disease and management course. Consequently, the use of self-reported QoL assessment has become a valuable tool in both clinical practice and research. 

Esophageal cancer is one of the most aggressive cancers worldwide. Several studies have been designed to assess QoL, including different surgical techniques, different chemotherapy regimens, long-term QoL follow-up after esophagectomy, etc. Yang et al. showed that different surgical procedures of esophagectomy in patients with esophageal cancer had similar good long-term QoL, including global health status and functional and symptom scales [2]. Another study reported by Schandl revealed esophageal cancer patients who experienced surgery had persistent reduced-health-related QoL compared to a population-based reference population according to quantified questionnaires for up to 10-year follow-up [3]. This finding was also reported in a Swedish study, which revealed that eating difficulties were still persistent for a duration of 10 years and were related to worse QoL in physical, role, social, and symptom functions, such as fatigue, insomnia, diarrhea, and appetite loss [4]. The QoL for different chemotherapy regimens has been also investigated. A French study demonstrated no significant differences in the analysis of health-related QoL in inoperable esophageal cancer patients between those who received oxaliplatin/leucovorin/fluorouracil (FOLFOX) versus those receiving cisplatin/fluorouracil [5]. Other assessments of QoL with different treatment modalities have also been estimated, such as surgery with or without concurrent chemoradiotherapy and different surgical approaches to esophagectomy [6,7].

Recently, there has been a worldwide tendency to decrease medical costs and a medical insurance revolution. In this background, outpatient chemotherapy (OPCT) has become a choice of therapeutic modality. Compared to traditional inpatient chemotherapy (IPCT), OPCT has several benefits, such as shortening hospital stays, cutting the cost of admission, and effectiveness of hospital beds. Growing evidence has demonstrated the safety and efficacy of OPCT in several cancer types [8,9,10,11]. A Japanese study revealed the equal quality of life in IPCT and OPCT; however, it included patients with different cancer types and different chemotherapy regimens [10]. Therefore, it is important to investigate the QoL of IPCT and OPCT in the same cancer population with the same chemotherapy regimen. 

Esophageal squamous cell carcinoma (ESCC) is the ninth leading cause of cancer-related deaths in Taiwan [12]. The majority of ESCC cases in patients are locally advanced disease on diagnosis; hence, chemotherapy is one of the standard therapies for these patients, whether alone or in combination with radiotherapy. Most patients with esophageal cancer have received IPCT. With the improvement of palliative care, such as antiemetic medication and analgesics, the side effects of chemotherapy have significantly decreased recently. Additionally, because the incidence of cancer is becoming higher, the administration of chemotherapy has steadily increased. Recently, a new device, the elastomeric ambulatory pump (Baxter Corporation, Mississauga, Ontario, Canada) has been developed. This device must be used through central venous access, such as a port catheter, and is single-use only. Because of the low infection risk, portable use, easy performance, low bottle volume, and lack of admission, an increasing number of patients undergo chemotherapy in the outpatient clinic. However, to the best of our knowledge, there is limited evidence concerning the differences in QoL and side effects between patients receiving IPCT and those receiving OPCT. Hence, this study aimed to compare the QoL and safety between these two groups.

## 2. Materials and Methods 

### 2.1. Patient Selection 

Patients aged >20 years were eligible for inclusion in this study. In addition, only squamous cell carcinoma was allowed; moreover, patients need to be willing to participate and able to express themselves well orally or in writing in Chinese. Between September 2015 and August 2017, 107 ESCC patients who underwent chemotherapy with cisplatin/5-fluouracil during admission or outpatient clinic at Kaohsiung Chang Gung Memorial Hospital were enrolled. These patients all met the following criteria: (1) no major surgery in the last 6 months; (2) Eastern Cooperative Oncology Group (ECOG) performance status 0–1; (3) no evidence of brain or leptomeningeal metastasis; (4) no radiotherapy performed in the past two months, whether concurrent or sequential with chemotherapy; (5) no body weight loss >5% per week; (6) albumin level >3.0 g/dL; (7) no current infection or inflammation status requiring antibiotic treatment; (8) no history of second primary malignancy. Each patient had to complete at least three cycles of chemotherapy after enrollment. The 8th American Joint Committee on Cancer (AJCC) staging system was used to determine tumor stage for each ESCC patient [13].

### 2.2. Quality of Life Assessment

The European Organization for Research and Treatment of Cancer Quality of Life Questionnaire Core 30 Items (EORTC QLQ-C30), the most validated QoL tool in oncology, is composed of a multi-item scale that reflects the multidimensionality of the QoL construct [14]. It incorporates five functional scales, physical (5 questions), role (2 questions), cognitive (2 questions), emotional (4 questions), and social (2 questions); three symptom scales, nausea (2 questions), pain (2 questions), and fatigue (3 questions); 6 other single questions to investigate the severity of symptoms, including insomnia, dyspnea, diarrhea, constipation, loss of appetite, and financial problem; and two final questions for the global health assessment. According to the EORTC QLQ-C30 scoring manual, scores of each scale were calculated within a range of 0 to 100 [15]. A higher coefficient for the functional scales corresponds to a better level of functioning, but a higher score on the symptom scale indicates worse QoL or more problems. This questionnaire is used for general health assessment as well as physical, emotional, and social assessment. The EORTC QLQ Oesophageal Cancer Module (EORTC QLQ-OES18) is an esophageal site-specific module designed to collect information about disease as well as treatment-related symptoms and side effects [16,17]. This questionnaire is composed of four scales of disease-related symptoms—reflux, dysphagia, eating, and pain—and other six single scales of treatment-related side effects—choking, dry mouth, taste, cough, trouble with saliva swallowing, and speech. All scores ranged from 0 to 100, and a higher score means a greater escalation of the problem. All questionnaires have been translated into Chinese. The purpose of these questionnaires was explained to each patient in individualized interviews, and informed consent was delivered at the same time. The QoL questionnaire, EORTC QLQ-C30, and EORTC QLQ-OES18 were examined one week after chemotherapy, and the scores of these items were collected for each patient at our outpatient clinic.

The adverse events (AEs) of chemotherapy were evaluated according to the National Cancer Institute Common Terminology Criteria for adverse events (CTCAE) version 4.0, and the worst grade for each AE was recorded [18]. The document of AEs was performed according to electronic medical records and patients’ oral presentation one week after chemotherapy for each patient at our outpatient clinic.

### 2.3. Chemotherapy

Chemotherapy was administered in both groups, and the dose of chemotherapy was the same. Chemotherapy consisted of cisplatin 100 mg/m^2^ as 4-hour intravenous infusion and 5-fluorouracil 1000 mg/m^2^, from day 1 to day 4, as intravenous continuous infusion for 4 days every 3–4 weeks.

In the admission chemotherapy group, patients received chemotherapy intravenously and continuously, with cisplatin first, followed by 5-fluorouracil, from day 1 to day 4. In the OPCT group, they underwent cisplatin and the first pump (continuous 5-fluorouracil for 2 days) on day 1 and returned to our outpatient clinic for a second pump (continuous 5-fluorouracil for another 2 days) on day 3. After completion of chemotherapy, they returned to our hospital to remove the pump on day 5.

The option of IPCT or OPCT was dependent on patients’ request, based on convenience of OPCT, restriction of IPCT, health insurance, distance to our hospital, etc.

### 2.4. Statistics 

Statistical analyses were performed using the SPSS 19 software package (IBM, Armonk, NY, USA). The chi-square test and Mann-Whitney U test were used to compare the difference between the two groups. We used two-sided tests of significance, and *p* < 0.05 was regarded as significant.

### 2.5. Statement of Ethics 

The study was conducted in accordance with the declaration of Helsinki. The study was approved by the Institutional Review Board of Chang Gung Medical Foundation (104-4328B). Written informed consent was obtained from each participant and/or their legal representative.

## 3. Results

### 3.1. Patient Characteristics

A total of 107 ESCC patients who received chemotherapy were analyzed in our study, including 53 patients in the IPCT group and 54 patients in the OPCT group. Most patients were men, and the median age was 58 years. Almost 85% were stage III or IV, and more than 90% of patients had negative health behaviors such as cigarette smoking, alcohol drinking, and betel nut chewing. There were no significant differences between the two groups in terms of body height, body weight, age, gender, tumor stage, or personal history of smoking, alcohol, or betel nut use. The distribution of clinicopathological features in the two groups is shown in Table 1.

### 3.2. QLQ-C30

The qualified health-related QoL of each ESCC patient was assessed according to the EORTC QLQ-C30, and other conditions that may affect QoL were excluded, such as current infection, recent radiotherapy, poor ECOG status, and malnutrition. The results showed that the mean global QoL scores were similar between patients who received IPCT and those who received OPCT, as were functional scales, including physical, role, emotional, cognitive, and social function; although worse role (*p* = 0.09) and emotional (*p* = 0.05) problems with marginal significance were mentioned in the OPCT group. Regarding symptom scales, there were no statistical differences between these two groups. The clinically relevant differences in terms of insomnia, shortness of breath, pain, constipation, nausea, vomiting, poor appetite, and financial difficulties were similar, although there were more fatigue problems (*p* = 0.08) in the IPCT group. The distribution of QLQ-C30 parameters between the two groups is shown in Table 2.

### 3.3. QLQ-OES18

The results of EORTC QLQ-OES18 analysis revealed that the degree of dysphagia was similar, and there was no significant difference in the functional scales between the two groups. The IPCT group was found to have a borderline significant trend in eating (*p* = 0.07), reflux (*p* = 0.09), and dry mouth (*p* = 0.06) problems in comparison with the OPCT group. Other symptom scales, including pain, swallowing saliva, choking, taste, cough, and talking, were similar without any significant difference. The results of QLQ-OES18 analyses are summarized in Table 3.

### 3.4. AEs of Chemotherapy

The AEs associated with chemotherapy administration were recorded for each patient. Most AEs were grade 1–2; grade 3–4 toxicity was rare, including constipation (1.9%) in the OPCT group and grade 3 (1.9%) and grade 4 (1.8%) anemia in the IPCT and OPCT groups, respectively. There were no statistically significant differences in nephrotoxicity between these two groups, neither in frequency nor severity. Most patients tolerated the side effects of chemotherapy, and no patients experienced treatment-related deaths. There was no significant difference in AEs between the two groups. The profiles of these AEs are shown in Table 4. 

## 4. Discussion

QoL is an important issue for cancer patients who receive treatments such as surgery, chemotherapy, radiotherapy, or combination therapy. Chemotherapy is one of the most aggressive modalities of cancer treatment; it may contribute to increased response rate, prolonged survival, and potentially curative disease, but more AEs and symptoms, including nausea and vomiting, fatigue, and hematological toxicities. For esophageal cancer patients who underwent esophagectomy, adjuvant chemotherapy may bring about better global health status and physical, emotional, cognitive, role, and social functions in comparison with those without adjuvant chemotherapy, but more symptoms such as fatigue and dyspnea have also been mentioned [19]. 

In the past, most cancer patients received chemotherapy in the hospital, regardless of the chemotherapy regimen, the infusion time of chemotherapy, or the ECOG status. Recently, there has been an improvement in devices and health care systems; OPCT has become accepted and promoted, and an increasing number of patients undergo OPCT for cancer treatment, especially in Taiwan [16,17,18]. For example, cisplatin-5-fluorouracil, the standard care of chemotherapy for ESCC, was prescribed for them in the hospital with a duration of at least 4 days; however, more than half of the ESCC patients received the same regimen at the outpatient clinic in our hospital. The transition from IPCT to OPCT may increase the utilization of beds in hospitals, improve financial metrics and cost savings, decrease the cost of health insurance, and maintain acceptable AEs [20,21,22,23,24]. Nevertheless, there is still a disadvantage in OPCT: thromboembolism was reported to be a leading cause of cancer-related death in patients receiving OPCT [25]. However, the incidence of thromboembolism is relatively low in ESCC, and the ranking of ESCC is 15th out of 18 malignancies according to Medicare claims data [26]. In our study, the results showed no significant difference in QoL or AEs between IPCT and OPCT, indicating that it is reasonable and safe for ESCC patients.

There were several limitations to our study. First, the sample size was relatively small, and we included only patients treated at a single institution. Second, the time of chemotherapy intervention varied for each patient, such that the survival outcome was difficult to evaluate. Third, the data were collected only after chemotherapy; thus, it was difficult to compare the changes in QoL from baseline between these two groups. However, to the best of our knowledge, this was the first study to investigate the QoL in ESCC patients who underwent IPCT or OPCT. Large, population-based cohort studies are warranted to validate the results of this study.

## 5. Conclusions

Our study suggests that health-related QoL and AEs were similar between patients with ESCC who received IPCT or OPCT. A large population-based cohort or a prospectively designed study is warranted to validate this finding.

## Figures and Tables

**Table 1 healthcare-08-00447-t001:** Characteristics of the 107 patients with esophageal squamous cell carcinoma who received inpatient or outpatient chemotherapy.

Characteristics	Inpatient Chemotherapy Group (*N* = 53)	Outpatient Chemotherapy Group (*N* = 54)	*p*-Value
Body height (cm) (median, range)	167.3(154.1–175.6)	165.5 (142.6.1–180.3)	
Body weight (kg) (median, range)	58.5 (39.8–77.5)	58.2 (40.8–94.3)	
Age (years)			
<60 years	39 (73.6%)	30 (55.6%)	0.05
≥60 years	14 (26.4%)	24 (44.4%)	
Sex			
Male	52 (98.1%)	50 (92.6%)	0.18
Female	1 (1.9%)	4 (7.4%)	
Tobacco smoking			
Yes	49 (92.5%)	48 (88.9%)	0.53
No	4 (7.5%)	6 (11.1%)	
Alcohol consumption			
Yes	49 (92.5%)	50 (92.6%)	0.98
No	4 (7.5%)	4 (7.4%)	
Betel nut chewing			
Yes	40 (75.5%)	40 (74.1%)	0.87
No	13 (24.5%)	14 (25.9%)	
Tumor stage			
II	8 (15.1%)	9 (16.7%)	0.96
III	22 (41.5%)	22 (40.7%)	
IV	23 (43.4%)	23 (42.6%)	

**Table 2 healthcare-08-00447-t002:** EORTC-QLQ-C30 symptom scores.

Characteristics	Inpatient Chemotherapy Group (*N* = 53) Median (range)	Outpatient Chemotherapy Group (*N* = 54) Median (range)	*p*-Value
Global health status			
Global health status/quality of life	50 (0–100)	54 (0–100)	0.53
Functional scales			
Physical functioning	87 (73–100)	93 (73–100)	0.27
Role functioning	83 (33–100)	100 (33–100)	0.09
Emotional functioning	83 (8–100)	96 (25–100)	0.05
Cognitive functioning	83 (33–100)	83 (33–100)	0.38
Social functioning	100 (0–100)	100 (33–100)	0.12
Symptom scales			
Fatigue	33 (0–100)	22 (0–100)	0.08
Nausea and vomiting	17 (0–67)	0 (0–100)	0.71
Pain	8 (0–83)	17 (0–83)	0.64
Symptom items			
Dyspnea	0 (0–100)	0 (0–100)	0.34
Insomnia	33 (0–67)	33 (0–100)	0.76
Appetite loss	33 (0–100)	33 (0–100)	0.17
Constipation	33 (0–67)	33 (0–100)	0.73
Diarrhea	0 (0–67)	0 (0–67)	0.08
Financial difficulties	33 (0–100)	0 (0–100)	0.17

**Table 3 healthcare-08-00447-t003:** EORTC-QLQ-OES18 symptom scores.

Characteristics	Inpatient Chemotherapy Group (*N* = 53) Median (range)	Outpatient Chemotherapy Group (*N* = 54) Median (range)	*p*-Value
Symptom scales			
Eating	25 (0–83)	8 (0–67)	0.07
Reflux	17 (0–83)	0 (0–67)	0.09
Pain	11 (0–89)	11 (0–67)	0.25
Trouble swallowing saliva (OESSV)	33 (0–100)	0 (0–100)	0.99
Choking when swallowing (OESCH)	0 (0–100)	0 (0–100)	0.99
Dry mouth (OESDM)	33 (0–100)	0 (0–100)	0.06
Trouble with taste (OESTA)	0 (0–100)	0 (0–100)	0.65
Trouble with coughing (OESCO)	33 (0–100)	33 (0–100)	0.97
Trouble talking (OESSP)	0 (0–100)	0 (0–100)	0.27
Functional scales			
Dysphagia	44 (0–100)	33 (0–100)	0.99

**Table 4 healthcare-08-00447-t004:** Comparison of adverse events in 107 patients with esophageal squamous cell carcinoma who received inpatient or outpatient chemotherapy.

Characteristics	Inpatient Chemotherapy Group (*N* = 53)	Outpatient Chemotherapy Group (*N* = 54)	*p*-Value
Nausea			
Grade 0	43 (81.1%)	43 (79.6%)	0.80
Grade 1	7 (13.2%)	9 (16.7%)	
Grade 2	3 (5.7%)	2 (3.7%)	
Vomiting			
Grade 0	49 (92.4%)	50 (92.6%)	0.56
Grade 1	3 (5.7%)	4 (7.4%)	
Grade 2	1 (1.9%)	0 (0%)	
Mucositis			
Grade 0	45 (85.0%)	47 (87.0%)	0.67
Grade 1	4 (7.5%)	2 (3.7%)	
Grade 2	4 (7.5%)	5 (9.3%)	
Diarrhea			
Grade 0	45 (85.0%)	49 (90.7%)	0.29
Grade 1	8 (15.0%)	4 (7.4%)	
Grade 2	0 (0%)	1 (1.9%)	
Constipation			
Grade 0	30 (56.6%)	28 (51.9%)	0.25
Grade 1	11 (20.8%)	18 (33.3%)	
Grade 2	12 (22.6%)	7 (12.9%)	
Grade 3	0 (0%)	1 (1.9%)	
Neutropenia			
Grade 0	26 (49.1%)	26 (48.1%)	0.61
Grade 1	27 (50.9%)	27 (50.0%)	
Grade 2	0 (0%)	1 (1.9%)	
Anemia			
Grade 0	19 (35.8%)	19 (35.2%)	0.70
Grade 1	22 (41.5%)	21 (38.9%)	
Grade 2	11 (20.8%)	13 (24.1%)	
Grade 3	1 (1.9%)	0 (0%)	
Grade 4	0 (0%)	1 (1.8%)	
Thrombocytopenia			
Grade 0	36 (67.9%)	37 (68.5%)	0.99
Grade 1	16 (30.2%)	16 (29.6%)	
Grade 2	1 (1.9%)	1 (1.9%)	
Nephrotoxicity			
Grade 0	27 (50.9%)	23 (42.6%)	0.75
Grade 1	16 (28.3%)	19 (35.2%)	
Grade 2	7 (15.1%)	10 (18.5%)	
Grade 3	3 (5.7%)	2 (3.7%)

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
