# Peer review of "Similar Quality of Life and Safety in Patients Receiving Inpatient or Outpatient Chemotherapy: A Focus on Esophageal Squamous Cell Carcinoma"

_healthcare, 2020, doi:10.3390/healthcare8040447_

Round 1

Reviewer 1 Report

The paper reports on a comparison between self-reported quality of life for outpatient and inpatient groups treated for esophageal squamous cell carcinoma. The paper is well-structured and succinct; the methodology for the study including statistical analysis is clearly set out and the findings are generally well-presented. The main conclusion of the study, to the effect that there seems little difference in QoL ratings between the two groups strikes as noteworthy and significant for clinical practice, especially if, as the authors claim, this has not been researched before in the case of ESCC.

I note that clinical and other characteristics such as age and gender appear well-balanced between the two groups (Table 1). This would seem to lend support to the validity of the study's conclusions.. However I'd add that I am also thus left to wonder why some patients are in fact treated in hospital and others in OP - this is not an aspect the paper addresses.

Enlarging on this point somewhat, I feel a key deficiency of the paper as it stands is that  the 'Discussion' section and conclusion are somewhat sparse. I'd be interested to know, for example, more about the processes of clinical decision-making and/or patient choice that lead to either treatment route. Also, it might be instructive to bring to bear evidence related to the  distinction between outpatient/inpatient experience in terms of other cancers. At line 187 the authors note briefly moreover that thromboembolism was reported to be a leading cause of cancer-related death in patients receiving OPCT. This I think requires some expansion - e.g. why do they think this wasn't encountered in the study and how far is thromboembolism a specific risk with ESCC? 

I have one or two other fairly minor reservations in terms of the paper's content. The description of the EORTC QLQ-C30 (p3) seems somewhat muddled in that financial problems (l98) are grouped with physical symptoms. There also needs to be a little more clarity I feel about the significance of the use of pumps in the context of the Outpatient/Inpatient distinction. At l68 the ambulatory pump is described as a 'new device' that is used once, whereas two uses of pumps are mentioned in s2.3 (l121-23) in the context of OP treatment. Is either the same as the procedure mentioned earlier? And is use of a pump exclusive to the OP treatment regime, if so why?

Finally, while the paper is generally written clearly, there are a number of places where English language grammar and style requires attention. I am not able to list these exhaustively but there are examples in the Abstract at l30 (placement of 'either') and l34 (redundant 'a'). The sense of the reference to 'histology' at l78 is unclear and there is missing punctuation on l88. l148 'statistical' not 'statistically' etc. etc. Meanwhile, the reference to smoking, alcohol use and betel nut chewing as 'bad habits' (l136-137) is open to being read as rather pejorative: certainly these are widely regarded as negative health behaviours (and, as a non-clinician, I'm guessing particular risk factors in ESCC) but this passage could be rewritten to ensure that no judgment on patients' character is implied. 

Reviewer 2 Report

The authors evaluated the quality of life(QOL) and adverse events(AEs) in patients with esophageal squamous cell carcinoma who had received primary chemotherapy and found no difference in QOL or AEs between outpatient chemotherapy (OPCT) and inpatient chemotherapy (IPCT). The results suggest that changing chemotherapy to OPCT for esophageal cancer is feasible, but as the authors describe, the results need to be evaluated prospectively in the future. In terms of patient selection criteria, the study included patients with a good prognosis and may be considered evidence in a limited population.  Although there are no major problems with the objectives and results of the study, there are two points that warrant revision.

  1. Although patient selection is clearly described, the explanation of the study design and methods is unsatisfactory. It is difficult to understand from the description whether the patients' QOL was assessed by reminding them of their QOL during treatment and whether AEs were analyzed retrospectively based on their medical record information. Therefore, I would like the authors to add a detailed description of the study methodology. Also, it is unclear how the choice between IPCT and OPCT was made; was it a physician's suggestion or a patient's request? It is presumably the case that OPCT has increased over time, but it is also necessary to explain how the treatment was selected.

  1. One concern with cisplatin combination therapy in the outpatient setting is the risk of increased renal dysfunction due to inadequate control of water and urine volume. Even with adequate hydration, cisplatin nephropathy has a certain probability of occurring. Therefore, in this study, it is necessary to state whether there were AEs of renal dysfunction. Also, the authors should describe whether there was a difference in the frequency and severity of impaired renal function between inpatient and outpatient therapies. I believe that explaining this may alleviate concerns about taking outpatient chemotherapy.
